# Medication-Related Hospital Admissions and Emergency Department Visits in Older People with Diabetes: A Systematic Review

**DOI:** 10.3390/jcm13020530

**Published:** 2024-01-17

**Authors:** Azizah Vonna, Mohammed S. Salahudeen, Gregory M. Peterson

**Affiliations:** 1School of Pharmacy and Pharmacology, College of Health and Medicine, University of Tasmania, Hobart 7005, Australia; mohammed.salahudeen@utas.edu.au (M.S.S.); g.peterson@utas.edu.au (G.M.P.); 2Department of Pharmacy, Faculty of Mathematics and Natural Sciences, Universitas Syiah Kuala, Banda Aceh 23111, Aceh, Indonesia

**Keywords:** medication-related problems, adverse drug reaction, hospital admission, emergency department visit, diabetes mellitus, older people

## Abstract

Limited data are available regarding adverse drug reactions (ADRs) and medication-related hospitalisations or emergency department (ED) visits in older adults with diabetes, especially since the emergence of newer antidiabetic agents. This systematic review aimed to explore the nature of hospital admissions and ED visits that are medication-related in older adults with diabetes. The review was conducted according to the PRISMA guidelines. Studies in English that reported on older adults (mean age ≥ 60 years) with diabetes admitted to the hospital or presenting to ED due to medication-related problems and published between January 2000 and October 2023 were identified using Medline, Embase, and International Pharmaceutical Abstracts databases. Thirty-five studies were included. Medication-related hospital admissions and ED visits were all reported as episodes of hypoglycaemia and were most frequently associated with insulins and sulfonylureas. The studies indicated a decline in hypoglycaemia-related hospitalisations or ED presentations in older adults with diabetes since 2015. However, the associated medications remain the same. This finding suggests that older patients on insulin or secretagogue agents should be closely monitored to prevent potential adverse events, and newer agents should be used whenever clinically appropriate.

## 1. Introduction

As the global population is now living longer, the world is experiencing an increase in both the number and proportion of older people [1,2]. The World Health Organization reported that there were 1 billion people aged 60 years and older in 2019 [3]. This number is predicted to grow considerably to 1.4 billion by 2030, accounting for around 15% of the world’s population [3]. Notably, older adults are more likely to have physiological deterioration and chronic medical conditions, including diabetes mellitus [4,5,6].

The prevalence of diabetes among older adults has increased significantly in recent decades, affecting approximately 33% of this demographic group worldwide [3]. Older individuals with diabetes, whether newly diagnosed or long-standing, encounter an elevated risk of complications [4,6,7,8]. As a result, many of them are prescribed complex medication regimens, which may include potentially inappropriate medications (PIMs) and polypharmacy, typically defined as the use of five or more medicines [9]. In addition to this, age-related physiological changes further elevate the risk of adverse medication outcomes in this group. Impaired renal function, reduced cognitive function, and increased frailty are among the factors contributing to this heightened risk [6,7].

Treating older adults with diabetes presents numerous challenges linked to medication-related problems (MRPs), or “events or circumstances involving a patient’s drug treatment that actually, or potentially, interfere with the achievement of an optimal outcome” [10]. Such challenges encompass the risks of hypoglycaemia, neurocognitive decline, falls, and even mortality [7]. Currently, limited data are available regarding medication-related hospitalisations or ED visits in older individuals with diabetes. The two antidiabetic drug groups that have been commonly linked with hypoglycaemia in older adults are sulfonylureas and insulin [11,12]. However, with the emergence of newer antidiabetic agents, such as dipeptidyl peptidase-4 (DPP4) inhibitors and sodium-glucose cotransporter-2 (SGLT2) inhibitors, reported to be safer and more effective in older adults [13,14,15], the use of these older agents (e.g., sulfonylureas and insulin/analogues) has decreased [16,17,18,19]. As a result, the trends in MRPs that result in hospital admissions or ED visits might also have changed. This study, therefore, aimed to elucidate the trends and incidence of hospital admissions and ED visits in older patients with diabetes that are medication-related. Additionally, the research aimed to analyse the medications and other risk factors commonly associated with these hospital admissions or ED visits. 

## 2. Materials and Methods

This systematic review was conducted and reported according to the Preferred Reporting Items for Systematic Reviews and Meta-Analysis (PRISMA) guideline [20]. The review protocol was registered with the International Prospective Register of Systematic Reviews (PROSPERO: CRD42022353864).

### 2.1. Eligibility Criteria 

In this study, we reviewed original research that reported hospitalisations or ED visits associated with MRPs in older people with diabetes. The medication involved did not have to be explicitly used for diabetes. Original research studies were eligible for inclusion if they fulfilled the following criteria: studies of hospital admissions or ED presentations, with data including older patients (mean age of 60 years and above) with diabetes mellitus whose admission or presentation was due to a MRP. The MRP reported could relate to (but was not limited to) overtreatment, undertreatment, drug interactions, poor adherence, medication error or adverse drug reaction (ADR). Included were observational studies designed to collect data on real-world patients and studies drawing on existing patient registries, insurance databases, and electronic medical records.

Studies were excluded if they were not in English, were written reporting the outcome as MRPs that occurred subsequent to hospitalisation or ED presentation, were published as a thesis, conference abstract, protocol, review, commentary, case report or case series, were designed in a trial or intervention setting, had less than 100 participants, or had inclusion criteria that targeted patients with specific medication (e.g., only patients treated with certain insulins or sulfonylureas in combination with antibiotics).

### 2.2. Search Strategy

Comprehensive, systematic searches of Ovid Medline, Ovid Embase, and Ovid International Pharmaceutical Abstracts (IPA) were performed to locate relevant studies published between 1 January 2000 and 20 October 2023 (Appendix A). Subject headings and truncated search terms related to MRPs (e.g., side effect, ADR, medication error, polypharmacy), antidiabetic agents (e.g., sulfonylureas, insulin, DPP4 inhibitor), hospitalisation (e.g., hospital admission, ED presentation), and diabetes (e.g., diabetes mellitus, T1DM, T2DM) were combined. Type 1 diabetes mellitus was included, considering that an increased proportion of individuals with type 1 diabetes are living into the later decades of life [21]. From the search results, a citation analysis and hand search were performed using Google Scholar and Web of Science. A flowchart of the search strategy is depicted in Figure 1.

### 2.3. Data Extraction

The initial title screening was performed by AV, followed by abstract and full-text review by two reviewers independently (AV, GP/MS), using Covidence (www.covidence.org) (accessed from 27 March to 7 November 2023). Using the pre-specified inclusion/exclusion criteria, two independent reviewers had to agree on all inclusions and exclusions at the abstract and full-text stages. A third reviewer made the final decision when discrepancies could not be resolved between the first two reviewers. Data from published studies were extracted into a standardised format that described: study characteristics (e.g., study design, setting, country, number of patients, year), patients’ characteristics, types of MRPs identified, and risk factors associated with medication-related hospital admissions or ED presentations.

### 2.4. Quality Assessment 

Two reviewers (AV, GP/MS) independently evaluated each study’s quality using the Joanna Briggs Institute (JBI) list, which has tools for different types of studies (prevalence, analytical cross-sectional, case-control, cohort studies). Discrepancies in judgment were solved by discussion between reviewers to reach a consensus. The items covered by each JBI tool are shown in Appendix A. No studies were excluded based on the quality assessment outcome. 

## 3. Results

### 3.1. Overview of Included Studies

The primary electronic literature search identified a total of 3966 articles in the three databases (Figure 1). Only 70 studies were included in the full-text review. Ultimately, 35 studies met the pre-defined inclusion criteria [22,23,24,25,26,27,28,29,30,31,32,33,34,35,36,37,38,39,40,41,42,43,44,45,46,47,48,49,50,51,52,53,54,55,56], of which 8 studies were identified through citation analysis. Studies were published between 2005 and 2023 and conducted mainly in the United States (US) (*n* = 7) [24,26,36,42,47,49,52], Italy (*n* = 6) [23,29,31,33,41,53], Korea (*n* = 4) [30,38,48,55], Denmark (*n* = 2) [45,46], Germany (*n* = 2) [25,43], Greece (*n* = 2) [22,28], Japan (*n* = 2) [37,56], Portugal (*n* = 2) [35,44], Taiwan (*n* = 2) [32,50], Malta (*n* = 1) [51], Pakistan (*n* = 1) [27], France (*n* = 1) [54], Spain (*n* = 1) [39], Thailand (*n* = 1) [40], and the United Kingdom (UK) (*n* = 1) [34].

### 3.2. Quality Assessment

In terms of study orientation, the majority were retrospective studies (*n* = 29), with study designs varying from analytical cross-sectional (*n* = 16) [22,24,29,30,31,33,35,40,43,48,50,51,52,53,55,56], cohort (*n* = 10) [25,26,32,34,36,41,42,46,47,49], prevalence (descriptive cross-sectional) (*n* = 6) [23,27,37,39,44,54], to case-control (*n* = 3) [28,38,45]. Most studies (60%) were rated as being high quality (Appendix A) [23,24,30,32,35,36,37,38,39,40,41,43,44,45,47,48,52,53,54,55,56].

### 3.3. Study and Patient Characteristics

The included studies used objective criteria obtained from medical records (*n* = 18) [22,23,27,28,29,31,33,35,36,37,39,44,47,49,50,51,53,54,56], administrative databases (*n* = 12) [24,26,30,32,38,42,43,45,46,48,55], patients’ registries (*n* = 4) [25,34,40,52] or a combination of sources (*n* = 1) [41] to identify patients, their characteristics, and the study outcomes. The outcomes observed, hospital admissions and/or ED visits, were predominantly recorded using the International Classification of Diseases 10th Revision (ICD-10) (*n* = 14) [30,34,36,42,43,45,46,47,48,49,52,54,55,56]. Almost all studies described the subtype of diabetes among their study participants, with the majority being type 2 (*n* = 22) and having had diabetes for 6.4 to 21 years. Medication-related hospital admissions and ED visits found in this review were all ADRs reported as episodes of hypoglycaemia (Table 1).

### 3.4. Trends and Incidence of Hospital Admissions and ED Visits

Of the 35 included studies, 25 (71.4%) reported on the events, while the remaining 10 (28.6%) reported on trends in hospitalisation or ED visits between 1996 and 2019 in patients with mean ages ranging from 60.5 ± 14.2 to 84.7 ± 4.3 years. The trends in medication-related hospitalisations or ED visits were reported as incidence (events per person-years) [43,45,46,48,49,52,55], annual percent change [34] and percentage [36,44]. Of the majority of studies reporting the trends as incidence in the type 2 diabetes population during the early 2000s [43,46,48,49,55], values ranged between 356 and 1100 events per 100,000 person-years. These numbers experienced an upward trajectory in the early 2010s, reaching approximately 650 to 1327 events per 100,000 person-years. However, from 2015 to 2019, there was an observed decline to a range of 360 to 1060 events per 100,000 person-years. One study in patients with end-stage kidney disease as a concomitant condition revealed a similar declining trend in incidence [52].

### 3.5. Factors Associated with Hospital Admissions and ED Visits

#### 3.5.1. Medications

Most studies (*n* = 23) employed multivariate analyses (Table 2 and Table 3) to find the association between medications or other suspected risk factors and hospitalisations/ED visits [25,26,28,30,31,33,35,36,38,40,41,42,43,46,47,48,49,50,52,53,55,56]. In studies focusing on antidiabetic medications, insulin and sulfonylureas were identified as significant contributors to hypoglycaemia-related hospitalisations or ED visits. The odds ratio (OR) for this association was as high as 13.92 (11.23–17.27) [30]. Additionally, several studies reported that metformin and newer antidiabetic agents (GLP-1 agonists, SGLT2 inhibitors) were associated with lower odds of hypoglycaemia-related hospitalisation [25,26,36,46,47,55]. A similar association was also reported for DPP4 inhibitors, although one study reported an increased risk with linagliptin, sitagliptin, teneligliptin and vildagliptin [56].

Two studies explored the link between non-antidiabetic medications and the risk of hypoglycaemia-related admissions [46,56]. One of the studies found significantly elevated odds for diazoxide (OR 15.49 [95% CI 4.87–49.31]), a high risk for methylphenidate (OR 5.15 [95% CI 1.53–17.28]) and disulfiram (OR 4.21 [95% CI 2.05–8.62]), and a slightly increased risk for antidepressants (OR 1.11 [95% CI 1.02–1.21]) [56]. However, the data regarding corticosteroids produced conflicting results. One study suggested a substantial elevation in risk (OR 2.03 [95% CI 1.70–2.44]) [56], while another presented a contradictory 22% decrease in risk (OR 0.78 [95% CI 0.74–0.83]) [46] of hospital admission. Opioids were reported to be associated with a 16% lower risk of hospitalisation (OR 0.84 [95% CI 0.80–0.88]) [46].

#### 3.5.2. Other Risk Factors

Other risk factors associated with hospital admissions or ED visits that were frequently reported included age, with older people being more at risk [28,30,33,40,42,46,56]. Gender was also reported as a contributing factor, with females tending to exhibit a higher risk [30,40,43,49,52,55]. Past episodes of hypoglycaemia [26,36,38,41,42,55], multiple comorbidities [36,38,48], and longer duration of diabetes were all associated with a greater risk [46,47]. People with pre-existing chronic conditions, such as cardiovascular disease [36,38,50], hypertension [40,55], kidney disease, and cognitive impairment (including dementia), demonstrated an increased likelihood of experiencing hypoglycaemia-related hospitalisations or ED visits [26,28,30,31,33,38,40,43,47,49,50,53].

## 4. Discussion

The objective of this review was to expand understanding of the overall trend and incidence of medication-related hospitalisation or ED presentation in older adults with diabetes and to identify medications or other factors associated with these events. Our review detected studies performed in multiple countries, except for countries in Africa, South America, and Oceania. One study from Australia underwent full-text screening but was eventually removed from the review due to the outcome of the study (ambulance calls due to hypoglycaemia) [57]. Therefore, more studies are still required to provide robust data across different geographic and cultural contexts. 

### 4.1. Interpretation of Findings in Relation to Other Studies

Other systematic review studies with different populations and conditions [58,59,60] listed various types of MRPs (e.g., ADRs, PIMs, drug-drug interaction) associated with hospital admission. However, our review in diabetes only discovered admissions or ED presentations due to hypoglycaemia episodes. The limited number of studies that explicitly mention medication or drug-related hospitalisation in older adults with diabetes raises concerns that perhaps the events are being under-reported or unnoticed by healthcare professionals. 

Most hypoglycaemia events identified in this review were based on diagnosis codes documented in administrative or electronic health record data. Compared to self-report, this source of data could minimise the bias of detecting hypoglycaemia that sometimes can be misinterpreted in the older adult due to their overlapping features with geriatric syndromes [2]. However, in many of the studies [24,27,30,33,36,38,42,43,48,49,51], there was no differentiation made between the outcomes of ED visits and hospitalisations. Consequently, for these studies, it remains unclear whether the ED visits ultimately led to hospitalisation or not. 

Overall, this review shows that hypoglycaemia-related hospitalisations or ED visits have declined since 2015. Although the trends in hospitalisations or ED visits seem to portray a similar pattern, there was considerable diversity in incidence rates between countries, with the lowest incidence rate (22 events per 100,000 people in 2017) being reported in Denmark [45]. Various studies that also observed the trend in antidiabetic drug use during the time found a decrease in the use of medications that have a high contribution to hypoglycaemia incidents [34,46,48]. This might explain the drop in hypoglycaemia events during the observation period. One study that assessed the trend of hospitalisation and the use of sulfonylureas indicated that the decreased use of the drugs might have contributed to the lower incidence of hospitalisation in recent years [34]. Two studies conducted in the US observed an increasing trend when they utilised a combination of ICD-9 and ICD-10 diagnosis codes to identify hypoglycaemia-related ED visits or hospitalisations [36,49]. However, it is important to note that these two studies employed different diagnostic codes (ICD-9 and ICD-10) to determine the outcomes, making it challenging to describe the observed trend over the study period conclusively. Additionally, some studies have suggested that using ICD-10 codes can increase the specificity in describing medical conditions [61,62], thus identifying and recording more hypoglycaemia cases. 

Despite the seemingly decreasing rate of hypoglycaemia-related hospitalisation, the older antidiabetic agents (insulin and sulfonylureas) were still reported to be associated with hypoglycaemia-related hospital admissions or ED visits in older adults. Insulin, whether in the form of analogues or human, basal or mixed, was commonly reported (14 studies) to be associated with admission or ED presentation in older adults with diabetes [26,28,29,30,36,38,41,42,43,47,48,49,53,55]. The updated 2019 Beers Criteria only listed insulin given in a sliding-scale dosage as a PIM for older adults. However, our review could not confirm whether the patients in the studies above were receiving sliding-scale doses. Another antidiabetic class linked to hypoglycaemia-related admissions are sulfonylureas, with the risk reported to be up to 13 times higher in one study [30]. The 2019 Beers Criteria updated its recommendation in regard to the use of sulfonylureas from avoiding only long-acting sulfonylureas to avoiding all sulfonylureas in older adults. If considering a sulfonylurea for older adult patients, several guidelines suggest opting for a short-acting one, such as glipizide [63,64]. A multitude of factors complicate the management of diabetes in older adults [6]. The patient’s overall health status, coexisting illnesses, and degree of cognitive function must be thoroughly considered prior to starting any form of glucose-lowering therapy for the older adult [7]. These findings further emphasise the need for extreme caution and close monitoring when prescribing insulin and secretagogue agents to older adults. 

In this review, we found that metformin and SGLT2 inhibitors were consistently linked to decreased odds of hospitalisation or ED visits due to hypoglycaemia [26,36,46], presumably because of their different mechanisms of action. However, more observational real-world data are required to comprehensively assess the risk of hypoglycaemia from the newer drugs. Clinical trials have reported severe hypoglycaemia episodes occurring in 0.7% to 2.4% of patients treated with newer antidiabetic medications (SGLT2 inhibitors, GLP-1 analogues, gliptins) [2]. Considering the nature of clinical trials, it is suspected that the rate of events in the real world could be higher. Among non-antidiabetic medications associated with an increased risk of hospitalisation or ED visits due to hypoglycaemia, diazoxide exhibited the highest risk [46]. Since diazoxide is indicated to counter hypoglycaemia in conditions like insulinoma and congenital hyperinsulinism [65], it is possible that patients with these conditions had pre-existing hypoglycaemia and were treated with diazoxide before hospital admission.

In terms of gender, females were found to be at higher risk of experiencing hypoglycemia-related hospitalisation or ED visits [30,40,43,49,55]. Several findings from different studies may shed light on this association. For instance, women tend to have reduced intrinsic counter-regulatory responses to hypoglycaemia [66]. Several research studies have consistently pointed out an increased likelihood of subsequent hypoglycaemia-related hospitalisations in individuals who have previously experienced such events [26,36,38,41,42,55]. This highlights the urgent importance of conducting thorough medication reviews and providing comprehensive patient education for those who have experienced hypoglycaemia-related hospitalisation or ED visits. 

Among significant comorbidities reported to elevate the odds of hypoglycaemia-related hospitalisations and ED visits were cardiovascular disease, hypertension, dementia or cognitive impairment, and chronic kidney disease [28,38,40]. Diabetes, hypertension and cardiovascular disease are closely intertwined due to the sharing of risk factors, such as endothelial dysfunction, arterial remodelling, vascular fibrosis, atherosclerosis, dyslipidemia, and obesity [67]. Individuals with diabetes who have normal blood pressure tend to exhibit reduced insulin secretion, resulting in a lower risk of hypoglycemia [68].

A previous study has reported an elevated risk of hypoglycaemia in older adult patients with diabetes who also have dementia [24]. Older adults with dementia often have a reduced dietary intake, a decline in cognitive and functional capacity, and a heightened likelihood of hypoglycaemic unawareness [69]. These characteristics can mask hypoglycemia symptoms, potentially leading to the development of more severe hypoglycemia episodes requiring hospital admission or an ED visit. Kidney disease, whether in the form of a reduced estimated glomerular filtration rate (eGFR) or chronic kidney disease, has consistently been reported to elevate the odds of hypoglycaemia-related hospitalisations or ED visits [26,28,30,31,33,38,40,43,47,49,50,53]. Impaired kidney function affects the clearance of many antidiabetic medications, which in turn prolongs the drugs’ presence in the body, subsequently increasing the risk of hypoglycaemia [6]. 

### 4.2. Strength and Limitations

The diversity of clinical settings and how the incidences were reported from country to country made it difficult to compare the trends. MRPs identified in our studies were exclusively hypoglycaemia as ADRs. Taking into account that our review only included studies that reported MRPs that lead to hospital admission or ED presentations, it can be said that the hypoglycaemia reported in this review accounted for the most severe events. A true rate of hypoglycaemia that was self-reported or occurring at home was unavailable from this study, and, therefore, the review’s findings might only represent a small portion of all ADR events associated with antidiabetic drugs [2].

### 4.3. Conclusions

This systematic review highlights a notable gap in the literature, with a relative lack of studies explicitly addressing MRPs that may result in hospital admissions or ED visits. All identified studies focussed on hypoglycaemia. While the frequency of hypoglycaemic events leading to hospitalisation in older adults with diabetes seems to have decreased, the medications associated with such hospitalisations remain consistent: insulin and sulfonylureas. This finding emphasises the importance of monitoring older diabetes patients on insulin and secretagogues to prevent potential adverse events. Moreover, it highlights the need to consider newer agents when appropriate.

## Figures and Tables

**Figure 1 jcm-13-00530-f001:**
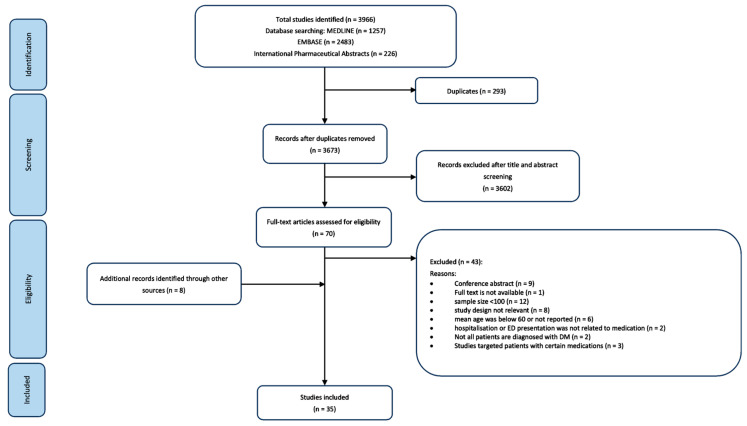
PRISMA flow diagram of the study selection process and citation analysis.

**Table 1 jcm-13-00530-t001:** Study and participant characteristics, trends or events of medication-related hospital admissions and emergency department (ED) visits in older adult patients with diabetes.

Author, Year, Country	Study Design	Data Sources	Years Reported (Duration) (Years)	Type of Diabetes	Patients’ Age (Mean [SD]/Median [IQR]) (Years)	Outcome Observed (Diagnosis Code)	Trend or Events of Outcome Observed
**Studies that reported medication-related hospital admissions and/or emergency department (ED) visits as *the trend over time***							
Zhong et al. (2017), UK [34]	retrospective, cohort	patients’ registry	1998–2013 (16)	T2DM	100% ≥65 years	hospitalisations due to hypoglycaemia (ICD-10)	(Annual percent change [95% CI]) Decreased: 1998–2009: 8.59 [5.76–11.50]; 2009–2013: −8.05 [−14.48 to −1.13] of older adult patients with T2DM.
Misra-Hebert et al. (2018), US [36]	retrospective, cohort	hospital medical record	2006–2015 (10)	T2DM	61.0 (51.9, 69.9)	ED visits or hospitalisations due to hypoglycaemia (ICD-9/-10)	(Percentage [%]) Increased: 2006–2015: 0.12 to 0.31 (*p* = 0.01) of patients with T2DM who had either a primary care or endocrinology visit within 2 years.
Müller et al. (2020), Germany [43]	retrospective, cohort	administrative database	2006, 2011, 2016 (1)	T2DM	2006: 72.8 (12.4); 2011: 73.4 (12.3); 2016: 73.0 (13.2)	ED visits or hospitalisations due to hypoglycaemia (ICD-10)	(Events per 100,000 people) Decreased: 2006 = 460; 2011: 490; 2016: 360 of patients with T2DM.
Pereira et al. (2020), Portugal [44]	retrospective, prevalence	hospital medical record	2012–2016 (5)	T1DM and T2DM	71 (57–81)	ED visits due to hypoglycaemia (ICD-9)	(Percentage [%]) Decreased: 0.15% in 2012 to 0.10% in 2016 (*p* < 0.001) of all ED visits.
Bengtsen et al. (2021), Denmark [45]	retrospective, case-control	administrative database	1997–2017 (21)	T1DM and T2DM	73.70 (63.60, 81.50)	hospitalisations due to hypoglycaemia (ICD-10)	(Events per 100,000 people) Decreased: 1997–2003: 17.7–30.3; 2010: 30.4; 2017: 22.0 of patients with diabetes.
Jensen et al. (2021), Denmark [46]	retrospective, cohort	administrative database	1998–2018 (21)	T2DM	61 (17)	hospitalisations due to hypoglycaemia (ICD-10)	(Events per 100,000 people) Decreased: 1998: 700; 2003: 1100; 2018: 400 of patients with T2DM.
Lee et al. (2021), Korea [48]	retrospective, analytical cross-sectional	administrative database	2006–2015 (10)	T2DM	100% ≥65 years	ED visits or hospitalisations due to hypoglycaemia (ICD-10)	(Events per 100,000 people) Decreased: 2006–2010: 859 to 1327, 2011–2015: 1262 to 1060 of T2DM people aged 65 years or older.
Pilla et al. (2021), US [49]	retrospective, cohort	hospital medical record	2009–2019 (11)	T2DM	61.0 (14.3); 47% ≥65 years in 2014	ED visits or hospitalisations due to hypoglycaemia (ICD-9 and ICD-10)	(Events per 100,000 people) Decreased: 2009: 270; 2014: 160 (ICD-9). Increased: 2016: 560, 2019: 660 (limited ICD-10); 2016: 630, 2019: 730 (expanded ICD-10) of patients with T2DM who had either a primary care or endocrinology visit within 3 years.
Galindo et al. (2022), US [52] *	retrospective, cohort	patients’ registry	2013–2017 (5)	not specified	65.0 (57.0–73.0)	ED visits or hospitalisations due to hypoglycaemia (ICD-9 and ICD-10)	(Events per 100,000 people) Decreased: 2013: 6400; 2017:4750
Yun et al. (2022), Korea [55]	retrospective, cohort	administrative database	2002–2019 (18)	T2DM	more than 50% are people ≥ 65 years	ED visits due to hypoglycaemia (ICD-10)	(Events per 100,000 people) Decreased from 2013 to 2019 2003–2012: 356 to 684, 2013–2019: 650 to 443 of patients with T2DM.
**Studies that reported medication-related hospital admissions and/or emergency department (ED) visits as *events during time period***							
Sotiropoulos et al. (2005), Greece [22]	prospective, analytical, cross-sectional	hospital medical record	1996–1999 (3)	T2DM	62.1 (8.7)	hospitalisations due to hypoglycaemia (NA)	207 out of 2858 T2DM patients admitted to hospital (7.2%).
Greco et al. (2010), Italy [23]	prospective, prevalence	hospital medical record	2001–2008 (8)	T2DM	84.7 (4.3) (100% ≥80 years)	hospitalisations due to hypoglycaemia (NA)	99 patients out of 591 medical admissions due to diabetes in patients ≥80 years.
Feil et al. (2011), US [24]	Prospective, analytical, cross-sectional	administrative database	2002–2003 (1)	not specified	100% ≥65 years	ED visits or hospitalisations due to hypoglycaemia (ICD-9)	37,343 patients out of 497,900 veterans with diabetes mellitus aged 65 and older (7.5%).
Tschöpe et al. (2012), Germany [25]	prospective, cohort	patients’ registry	June 2009–March 2010 (1)	T2DM	66.8 (57.8–74.1)	hospitalisations due to hypoglycaemia (NA)	13 patients out of 3347 patients with T2DM (0.4%).
Fu et al. (2014), US [26]	retrospective, cohort	administrative database	2007–2010 (4)	T2DM	30.7% ≥65 years	hospitalisations due to hypoglycaemia (ICD-9)	0.006 per 1000 people in hospitalised T2DM patients.
Nazish et al. (2014), Pakistan [27]	prospective, prevalence	hospital medical record	2010–2013 (4)	not specified	60.46 (14.20)	ED visits or hospitalisations due to hypoglycaemia (NA)	118 patients.
Liatis et al. (2014), Greece [28]	prospective, case-control	hospital medical record	Not reported (1.375)	T2DM	76.7 (10.1)	ED visits due to hypoglycaemia (Whipple’s triad)	268 patients.
Salutini et al. (2015), Italy [29]	retrospective, analytical, cross-sectional	hospital medical record	2009–2013 (5)	T1DM and T2DM	71 (16)	ED visits due to hypoglycaemia (ICD-9)	500 episodes (401 patients).
Kim et al. (2016), Korea [30]	retrospective, analytical, cross-sectional	administrative database	2013 (1)	not specified	100% ≥65 years	ED visits or hospitalisations due to hypoglycaemia (ICD-10)	9.93 per 1000 people in patients with diabetes.
Mantovani et al. (2016), Italy [31]	retrospective, analytical, cross-sectional	hospital medical record	2010–2014 (5)	T2DM	75 (13)	ED visits due to hypoglycaemia (NA)	444 patients.
Hung et al. (2017), Taiwan [32]	retrospective, analytical, cross-sectional	administrative database	2001–2009 (9)	T2DM	70.1 (12.2)	ED visits or hospitalisations due to hypoglycaemia (ICD-9)	2588 out of 87,029 patients with T2DM (2.97%) in the one million Taiwan administrative database.
Mazzi et al. (2017), Italy [33]	retrospective, analytical, cross-sectional	hospital medical record	2011 (1)	T1DM and T2DM	71.5 (16.8)	ED visits or hospitalisations due to hypoglycaemia (NA)	1922 episodes of treated patients with diabetes.
Conceição et al. (2017), Portugal [35]	prospective, analytical, cross sectional	hospital medical record	Jan 2013–Jan 2014 (1)	T2DM	77.5 (45; 97)	ED visits due to hypoglycaemia (Whipple’s triad)	0.074% (95% CI 0.066–0.082) of all ED visits.
Namba et al. (2018), Japan [37]	retrospective, prevalence	hospital medical record	April 2014–March 2015 (5)	T1DM and T2DM	71.5 (58.0–81.0)	ED visits due to hypoglycaemia (NA)	2237 patients out of 346,939 patients with diabetes admitted to ED.
Park et al. (2018), Korea [38]	retrospective, cohort	administrative database	2011–2013 (3)	T2DM	60.79 (12.20)	ED visits or hospitalisations due to hypoglycaemia (KCD-7)	0.96% among pharmacologically treated patients with T2DM.
Caballero-Corchuelo et al. (2019), Spain [39]	retrospective, analytical, cross-sectional	hospital medical record	2012–2014 (3)	T2DM	75.4	ED visits due to hypoglycaemia (NA)	122 patients.
Kaewput et al. (2019), Thailand [40]	retrospective, analytical, cross-sectional	patients’ registry	2014 (1)	T2DM	100% ≥65 years	hospitalisations due to hypoglycaemia (NA)	356 patients out of 11,404 older adult patients with T2DM (3.1%).
Andreano et al. (2020), Italy [41]	retrospective, cohort	administrative database and hospital medical record	2015–2017 (3)	T1DM and T2DM	76.2% ≥65 years	ED visits due to hypoglycaemia (ICD-9)	2137 patients out of 168,285 residents recorded with DM and treated with antidiabetic medications (1.27%) (or 4.7 per 1000 patient-years).
McCoy et al. (2020), US [42]	retrospective, cohort	administrative database	2014–2016 (3)	T1DM and T2DM	65.8 (12.1)	ED visits or hospitalisations due to hypoglycaemia (ICD-9 and 10)	9.06 per 1000 people in patients with diabetes.
Lacy et al. (2021), US [47]	retrospective, cohort	hospital medical record	2012–2017 (6)	T2DM	60.9 (15.2)	ED visits or hospitalisations due to hypoglycaemia (ICD-10)	22.6 per 1000 people in treated patients with T2DM.
Chen et al. (2022), Taiwan [50] **	retrospective, cohort	hospital medical record	2001–2018 (18)	T2DM	77.5 (8.9)	ED visits (ICD-9)	494 patients out of 3877 patients with Alzheimer’s Dementia (AD) and with concomitant T2DM (12.74%)
Galea et al. (2022), Malta [51]	retrospective, cross-sectional	hospital medical record	2018 (1)	T1DM and T2DM	71.5 (15.5)	ED visits or hospitalisations due to hypoglycaemia (NA)	167 episodes out of 21,589 medical admissions (0.77%).
Nuzzo et al. (2022), Italy [53]	retrospective, analytical, cross-sectional	hospital medical record	2013–2017 (5)	T1DM and T2DM	75 (17)	ED visits due to hypoglycaemia (ICD-9)	302 patients.
Poret et al. (2022), France [54]	retrospective, prevalence	hospital medical record	2015–2018 (4)	T1DM and T2DM	68 (58–75)	ED visits or hospitalisations due to hypoglycaemia (ICD-10)	178 patients.
Horii et al. (2023), Japan [56]	retrospective, cross-sectional	administrative database	April 2014-October 2019 (5)	T2DM	70.4 (12.3)	hospitalisations due to hypoglycaemia (ICD-10)	10,376 patients out of 703,745 (1.47%).

ICD, International Classification of Diseases, T1DM, Type 1 Diabetes Mellitus, T2DM, Type 2 Diabetes Mellitus, NA: not available. * Included are patients with diabetes and end-stage kidney disease. ** Included are patients with Alzheimer’s dementia (AD) and with concomitant T2DM.

**Table 2 jcm-13-00530-t002:** Medications associated with hospital admission or ED presentation due to hypoglycaemia in older adult patients with diabetes.

Associated Factors	Increased Risk [Odd Ratio (OR)/Incidence Rate Ratios (IRR)/Hazard Ratio (95% CI)]	Decreased Risk [Odd Ratio (OR)/Incidence Rate Ratios (IRR) (95% CI)]
** *Medications* **		
Insulin analogues	14.40 (13.50–15.50) [43]	
Basal insulin	12.53 (8.90–17.64) [42]	
Basal insulin	23.21 (15.71–34.27) [42]	
Mixed insulin	13.50 (12.70–14.50) [43]	
	27.65 (20.32–37.63) [42]	
Human insulin	11.20 (10.50–12.00) [43]	
Insulin	2.13 (1.67–2.73) [49], 2.77 (1.98–3.89) [36]	
	7.44 (6.63–8.36) [38], 2.00 (1.31–3.05) [41]	
	6.59 (4.43–9.79) [25], 4.51 (3.49–5.83) [48]	
	1.73 (1.67–1.79) [55], 2.35 (1.42–3.95) [28]	
	4.20 (3.39–5.19) [26], 4.68 (3.84–5.71) [50]	
	13.92 (11.23–17.27) [30], 3.41 (1.69–6.86) [47]	
	3.44 (3.25–3.64) [56]	
		0.66 (0.50–0.88) [33]
Insulin and sulfonylureas	4.74 (3.67–6.06) [26], 15.09 (13.60–16.74) [38]	
Sulfonylureas	5.70 (5.30–6.10) [43], 2.49 (1.92–3.22) [36]	
	3.29 (2.61–4.14) [48], 6.73 (4.93–9.22) [42]	
	1.94 (1.53–2.47) [55], 4.00 (2.51–6.36) [28]	
	5.71 (2.92–11.17) [35] *, 3.94 (3.42–4.55) [26]	
	1.98 (1.79–2.18) [38], 13.92 (11.23–17.27) [30]	
	2.27 (2.18–2.37) [46]	
Non- Sulfonylureas secretagogues (Glinides)	2.23 (1.79–2.18) [38], 1.38 (1.22–156) [56]	
Thiazolidinediones	1.82 (1.27–2.61) [48], 1.58 (1.24–2.00) [55]	
	1.92 (1.35–2.74) [50]	
DPP-4 inhibitor		0.51 (0.35–0.74) [36], 0.97 (0.75–1.23) [26]
		0.44 (0.38–0.49) [46], 0.52 (0.26–1.03) [25]
DPP-4 inhibitor (linagliptin)	1.62 (1.55–1.69) [56]	
DPP-4 inhibitor (sitagliptin))	1.05 (1.01–1.10) [56]	
DPP-4 inhibitor (teneligliptin)	1.23 (1.16–1.31) [56]	
DPP-4 inhibitor (vildagliptin)	1.29 (1.23–1.36) [56]	
GLP-1 agonist	1.59 (1.44–1.77) [56]	
		0.23 (0.08–0.62) [36], 0.51 (0.44–0.58) [46]
		0.62 (0.36–0.99) [26]
Metformin		0.43 (0.32–0.58) [36], 0.69 (0.54–0.87) [55]
		0.39 (0.16–0.93) [47], 0.72 (0.69–0.72) [56]
SGLT2 inhibitor		0.43 (0.33–0.56) [46], 0.65 (0.58–0.74) [56]
** *Non-diabetic medications* **		
Diazoxide	15.49 (4.87–49.31) [56]	
Methylphenidate	5.15 (1.53–17.28) [56]	
Disulfiram	4.21 (2.05–8.62) [56]	
Corticosteroid	2.03 (1.70–2.44) [56]	
		0.78 (0.74–0.83) [46]
Antidepressants	1.11 (1.02–1.21) [46]	
Opioids		0.84 (0.80–0.88) [46]

* Analysed as Secretagogue-based regimen. DPP-4, Dipeptidyl peptidase-4; SGLT2, Sodium-glucose cotransporter-2; GLP-1, glucagon-like peptide 1.

**Table 3 jcm-13-00530-t003:** Other risk factors associated with hospital admission or ED presentation due to hypoglycaemia in older adult patients with diabetes.

Associated Factors	Increased Risk [Odd Ratio (OR)/Incidence Rate Ratios (IRR)/Hazard Ratio (95% CI)]	Decreased Risk [Odd Ratio (OR)/Incidence Rate Ratios (IRR) (95% CI)]
Gender (female)	1.13 (1.10–1.20) [43], 1.32 (1.08–1.61) [49]	
	1.05 (1.02–1.08) [55], 1.63 (1.04–2.56) [40]	
	1.12 (1.04–1.21) [30], 1.05 (1.01–1.10) [56]	
	1.09 (1.06–1.12) [52]	
Older age	1.58 (1.23–2.02) [42], 1.80 (1.37–2.35) [40]	
	1.30 (1.20–1.45) [28], 3.00 (2.64–3.41) [30]	
	1.03 (1.02–1.04) [33], 2.99 (2.71–3.30) [46]	
	1.1 (1.04–1.16) [56]	
Longer duration of diabetes	1.02 (1.01–1.03) [47], 2.45 (2.29–2.61) [46]	
Increased number of diabetes medications	1.56 (1.36–1.79) [36], 4.00 (2.87–5.58) [48]	1.10 (0.51–2.46) [53]
Duration of insulin use		0.62 (0.52–0.74) [41]
HbA1c < 6%	1.95 (1.44–2.65) [36], 2.00 (1.33–2.94) [49]	
	1.45 (1.12–1.87) [42]	
HbA1C (≥8.5)	1.49 (1.06–2.09) [40], 2.01 (1.31–3.07) [47]	
	1.56 (1.05–2.33) [49]	
Previous hypoglycaemia	3.01 (2.09–4.34) [36], 8.47 (8.16–8.80) [55]	
	6.60 (5.77–7.56) [42], 5.34 (3.93–7.26) [41]	
	3.30 (1.89–5.35) [26], 7.74 (6.82–8.79) [38]	
Lower BMI	1.61 (1.54–1.69) [56]	
Higher BMI		0.44 (0.33–0.60) [40], 0.96 (0.94–0.98) [36]
Higher number of comorbidities	1.74 (1.34–2.27) [28], 1.56 (1.41–1.73) [33]	
	1.79 (1.37–2.3) [47], 4.12 (3.07–5.51) [42]	
Increased Charlson comorbidity index	1.15 (1.10–1.21) [36], 2.76 (1.82–4.18) [48] 8.84 (6.85–11.40) [38]	
Cardiovascular disease	1.68 (1.24–2.28) [36], 1.20 (1.15–1.25) [38]	
	2.21 (1.71–284) [50]	
Hypertension	1.29 (1.25–1.34) [55], 1.63 (1.04–2.56) [40]	
Kidney disease	10.30 (10.10–10.60) [43], 2.86 (2.33–3.57) [49]	
	1.59 (1.12–2.11) [40], 2.96 (1.51–6.11) [53]	
	4.53 (2.59–7.95) [28], 1.38 (1.20–1.57) [26]	
	2.52 (2.26–2.82) [30], 2.03 (1.09–4.20) [31]	
	1.71 (1.62–1.81) [38], 2.96 (2.01–4.35) [47]	
	1.56 (1.22–1.98) [33], 2.98 2.46–3.62) [50]	
Cognitive impairment/dementia	2.57 (1.85–3.56) [49], 6.98 (1.80–26.98) [40]	
	1.93 (1.76–2.12) [30], 10.16 (3.40–30.36) [28]	
	1.45 (1.08–1.95) [33]	

BMI, Body mass index; HbA1C, haemoglobin A1.

## Data Availability

The data presented in this study are available on request from the corresponding author (a.vonna@utas.edu.au).

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
