# Peer review of "Medication-Related Hospital Admissions and Emergency Department Visits in Older People with Diabetes: A Systematic Review"

_jcm, 2024, doi:10.3390/jcm13020530_

Round 1
Reviewer 1 Report
Comments and Suggestions for Authors
Very timely and interesting systematic review concerning medication-related hospital admissions and emergency departments visits in older people with diabetes due to their diabetes medication.
It is no surprise that hypoglycemia is the most common cause for it.
With the rise of new classes of diabetes medications (SGLT2-inhibitors, GLP-1-receptor analogues and DPP4-inhibtors) which per se do not induce severe hypglycemia it is no wonder that incidence rates of these acute complications seem to declone in those economically developped states where these more expensive drugs can be used on a broader basis also in older subjects with diabetes mellitus who are prone to hypoglycemic episodes due to various reasons which are discussed in the paper extensively.
It is good to see that real-world evidence seems to reflect this expectations.
Nonetheless, all problems are not solved since nat alle new medicatiosn are appropriate in all older diabetic patients (i.e. GLP-1-RA with GI-side effects and maybe unwished loss of weight in elderly persons).
In quite a relevant number of diabetic patients in higher age additional insulin treatment is inevitable because of diminshed autonomous insulin secretion.
Also "short-acting" sulfonylureas like gliclazide are not completely obsolet, but are third- or fourth line "reserve drugs" also in elderly persons (please see/cite also recent ADA/EASD Consensus paper and ADA guideline publication 2024) where they seem necessary and appropritate on an individual decision. Also economic reasons may play a role.
Thus, the authors should state in the discussion that in general, sulfonylureas may not be considered as obsolet or malpractice in the individual patient and that insulin therapy is very often absolutely necessary in older diabetic diabetic pesrons to stabilize metabolism also in combination with GLT2-inhibtors with their risk of euglycemic ketoacidosis in the case of absolute or relative insulin deficency. Decisive for safety is patients´ and care-givers´ education and information, the choice of appropriate sulfonylureas and insulins - if clinically necessary - and a higher HbA1c goal wich is safe also for elderly patiemts in the light of the used diabetic medications.
These proposals for clinical use should be integrated in the discussion section, preferable in the final pragraphs.
Author Response
We are grateful for the reviewers and editor for the in-depth and constructive comments and suggestions. Here we presented our replies and summary of revisions made.

Reviewer 2 Report
Comments and Suggestions for Authors
Please see the attached document

Author Response

(The authors gave the same response as above.)
